# NameGuess: Column Name Expansion for Tabular Data

**Jiani Zhang**[*†], **Zhengyuan Shen**[*], **Balasubramaniam Srinivasan, Shen Wang,**
**Huzefa Rangwala, George Karypis**
Amazon Web Services
{zhajiani, donshen, srbalasu, shenwa, rhuzefa, gkarypis}@amazon.com

## Abstract

Recent advances in large language models have revolutionized many sectors, including the database industry. One common challenge when dealing with large volumes of tabular data is the pervasive use of abbreviated column names, which can negatively impact performance on various data search, access, and understanding tasks. To address this issue, we introduce a new task, called NAMEGUESS, to expand column names (used in database schema) as a natural language generation problem. We create a training dataset of 384K abbreviated-expanded column pairs using a new data fabrication method and a human-annotated evaluation benchmark that includes 9.2K examples from real-world tables. To tackle the complexities associated with polysemy and ambiguity in NAMEGUESS, we enhance autoregressive language models by conditioning on table content and column header names – yielding a fine-tuned model (with 2.7B parameters) that matches human performance. Furthermore, we conduct a comprehensive analysis (on multiple LLMs) to validate the effectiveness of table content in NAMEGUESS and identify promising future opportunities. Code has been made available at https://github.com/amazon-science/nameguess.

## 1 Introduction

Tabular data is widely used for storing and organizing information in web (Zhang and Balog, 2020) and enterprise applications (Leonard, 2011). One common practice when creating tables in databases is to use abbreviations for column headers due to character length limits in many standard database systems. For example, the maximum length for column names in an SQL database is 256 bytes, leading to the use of abbreviations such as "D_ID" for "Department ID" and "E_NAME" for "Employee

---
[*] These two authors contributed equally.
[†] Corresponding author.

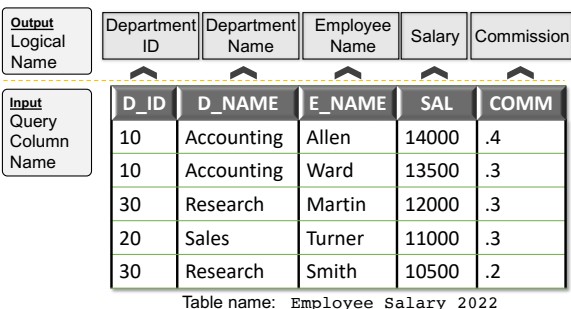

Figure 1: An example of the column name expansion task. The input are query column names with table context and the output are expanded logical names.

Name" as in Figure 1. While abbreviations can be convenient for representation and use in code, they can cause confusion, especially for those unfamiliar with the particular tables or subject matter. Column headers are essential for many table-related tasks (Xie et al., 2022), and using abbreviations makes it challenging for end users to search and retrieve relevant data for their tasks.

Abbreviated column names can negatively impact the usefulness of the underlying data. For example, in the text2SQL semantic parsing task, which converts natural language into formal programs or queries for retrieval, abbreviations can lead to a mismatch with the terms used in the natural language queries. In fact, in the human-labeled text2SQL *spider* dataset (Yu et al., 2018), 6.6% of column names are abbreviations. Figure 2 shows an example containing abbreviated column names like "c_name" and "acc_bal" in tables, which mismatch the terms "the name of all customers" and "account balance". Simple changes in using abbreviated column names in the spider dataset result in a performance degradation of over ten percentage points, with the exact match score of 66.63% (Xie et al., 2022) dropping to 56.09% on the T5-large model (Raffel et al., 2020). The effect of the abbreviated column names on table question answering (QA) (Yin et al., 2020), and column relation dis-

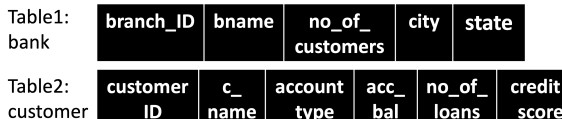

**Utterance** (input): List the name of all customers sorted by their account balance in ascending order.
**SQL** (output): SELECT ~~customer_ID~~ FROM customer ORDER BY ~~account_type~~

Figure 2: An example for Text2SQL semantic parsing. The terms "the name of all customers" and "account balance" do not match the abbreviated column names `c_name` and `acc_bal`. Instead, they match with the column names `customer_ID` and `account_type`.

covery ([Koutras et al., 2021](#)) are in Table 1 and description is in Appendix A.1. The performance degradation emphasizes the need for descriptive column headers in handling tabular data.

Expanding column names and generating descriptive headers also has other beneficial aspects. First, using expanded column names can increase the readability of tables, especially when complex or technical data is present. The expansion also enables data integration by allowing users to easily distinguish between tables with similar column names but different meanings and helping identify relationships between tables with different abbreviated column names. Finally, expanded column names can also improve the efficacy of keyword-based searches for discovering related tables.

This work addresses the task of expanding abbreviated column names in tabular data. To the best of our knowledge, this is the first work to introduce and tackle this problem. Unlike previous textual abbreviation expansion works that formulated the task as a classification problem with a predefined set of candidate expansions ([Roark and Sproat, 2014](#); [Gorman et al., 2021](#)), we formulate NAMEGUESS as a natural language generation problem. Acquiring extensive candidate expansions can be laborious, as pairs of abbreviated and expanded column names are seldom present in the same table. Conversely, abbreviation-expansion pairs can be gleaned from textual data through co-occurrence signals, such as parenthetical expressions. Moreover, abbreviated headers may exhibit ambiguity and polysemy arising from developer-specific naming conventions and domain-related variations in expansions.

To tackle NAMEGUESS, we first built a large dataset consisting of 163,474 tables with 384,333 column pairs and a human-annotated benchmark

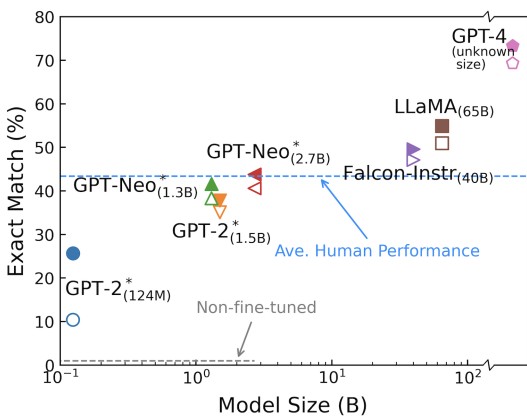

Figure 3: Exact match results for fine-tuned models (*), non-finetuned LLMs, and human performance. Solid and hollow symbols denote inclusion and exclusion of sampled table contents.

with 9,218 column pairs on 895 tables. We then proposed a method to produce training data by selectively abbreviating well-curated column names from web tables using abbreviation look-ups and probabilistic rules. Next, we enhanced autoregressive language models with supervised fine-tuning, conditioned on table content and column headers, and conducted extensive experiments to evaluate state-of-the-art LLMs. The overall model performance is shown in Figure 3. While GPT-4 exhibited promising performance on NAMEGUESS, the deployment of such LLMs comes with much larger memory and computation overheads. Our findings indicate that supervised fine-tuning of smaller 2.7B parameter models achieves close to human performance, and including table contents consistently improved performance. However, all models found the task challenging, with the best only achieving 54.7% accuracy on the extra-hard examples, indicating room for improvement in expanding abbreviated column names. Our main contributions are:

1. Introduced a new column name expansion task, named NAMEGUESS, as a natural language generation problem,
2. Developed a large-scale training dataset for the NAMEGUESS task using an automatic method that largely reduces human effort,
3. Created a human-annotated evaluation benchmark with various difficulty levels, which provides a standard for comparing results,
4. Performed a comprehensive evaluation of LMs of different sizes and training strategies and compared them to human performance on the NAMEGUESS task.

| | Original Column Names | Abbreviated Column Names |
|---|---|---|
| Text2SQL (Match score %) | 66.63 | 56.09 |
| Schema-based Relation Detection (Recall %) | 100.00 | 59.50 |
| Table QA (Accuracy %) | 84.32 | 80.49 |

Table 1: The effect of abbreviated column names on three table understanding tasks. The performance drops on all the tasks.

## 2 Problem Formulation

We formulate the NAMEGUESS task as a natural language generation problem: given a query name $x$ from table $t$, generate a logical name $y$ that describes the column. A table contains table content and various schema data, like a table name, column headers, and data types. Let $f_\theta$ be a generator with parameters $\theta$, the formulation becomes $y = f_\theta(x|t)$. Note that the query column names within a table may take different forms, including abbreviations and full names. The output logical names are expanded column names and should be easily understandable without additional knowledge of the table content. See Figure 1 for example inputs and outputs in the "Employee_Salary_2022" table, where "SAL" stands for "Salary" and "COMM" stands for "Commission".

## 3 Dataset Creation

We created a training dataset comprising 384,333 columns spanning 163,474 tables and a human-labeled evaluation benchmark containing 9,218 examples across 895 tables for NAMEGUESS. The main challenge in obtaining abbreviated-expanded column name pairs is the rare co-occurrence of these two names within the same table or database. Therefore, we employed the strategies of converting well-curated column names to abbreviated names that align with the naming convention of database developers and annotating the abbreviated column names based on the information in the input table. Figure 4 illustrates the main steps for creating the training and evaluation datasets. The details of the source tables are discussed in Section 3.1. The training and evaluation datasets are constructed in Section 3.2 and Section 3.3.

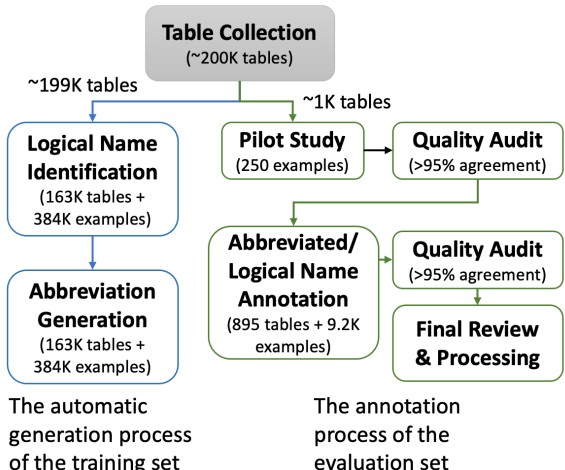

Figure 4: The processes of creating the training and evaluation datasets.

### 3.1 Table Collection

The training and evaluation datasets were obtained from seven public tabular dataset repositories. To ensure the quality of the tables, we filtered out tables with less than five rows or columns and removed tables with more than half of the entries being NaN or more than half of column names being duplicates. Table 2 summarizes the dataset statistics.

**City Open Data**. We sourced tables from New York (NYC), Chicago (CHI), San Francisco (SF), and Los Angeles (LA), covering categories, such as business, education, environment, health, art, and culture. We downloaded all the tables using Socrata Open Data APIs in June 2022.

**GitTables** was extracted from CSV files in open-source GitHub repositories (Hulsebos et al., 2023). GitTables is the largest dataset we can access with a relatively larger table size.

**WikiSQL** was released for the text2SQL task (Zhong et al., 2017). We only used the large corpus of Wikipedia tables in this dataset.

**Dresden Web Table Corpus** (Eberius et al., 2015). We only utilized the relational tables in this dataset and applied strict filtering criteria to keep tables with high-quality column names and contents. This is the largest accessible dataset but with relatively smaller tables than others.

### 3.2 Training Data Creation

We utilized two steps to convert logical names to abbreviated names: (1) identifying the logical names as the ground truth $y$ and (2) abbreviating the names as the input column names $x$.

| Data Source | #Ex. | #Table | Avg. #Col | Avg. #Row |
|---|---|---|---|---|
| Training Datasets | | | | |
| NYC | 16,697 | 1,921 | 23.2 | 642 |
| GitTables | 163,204 | 49,259 | 19.5 | 93 |
| WikiSQL | 22,963 | 9,268 | 6.4 | 20 |
| DWTC | 181,469 | 103,026 | 65.6 | 8 |
| **Overall** | 384,333 | 163,474 | 47.8 | 42 |
| Evaluation Datasets | | | | |
| SF | 4,781 | 388 | 23.9 | 643 |
| CHI | 3,975 | 442 | 21.1 | 605 |
| LA | 462 | 65 | 21.3 | 578 |
| **Overall** | 9,218 | 895 | 21.9 | 620 |

Table 2: Statistics for the training and evaluation datasets. '#Ex.' stands for 'number of examples'. 'Avg. #Col' is 'the average number of columns per table'.

### 3.2.1 Logical Name Identification

Identifying high-quality column names from relational tables is essential, as further abbreviating a vague term can lead to even more ambiguity. Algorithm 2 in Appendix A.2 shows the detailed algorithm with a vocabulary-based strategy. We regarded a column name as *well-curated* only if all the tokens in the column name can be found from a pre-defined vocabulary. Here, we used the *Word-Ninja* package to split the original column headers, which allowed us to check whether individual tokens are included in the vocabulary.

To construct this pre-defined vocabulary, we used WordNet open-class English words (Fellbaum, 1998) followed by a set of filtering criteria (such as removing words with digits, punctuation, and short words that are abbreviations or acronyms) so that the classifier achieves high precision for detecting the logical names, which further used as ground truth labels.

### 3.2.2 Abbreviation Generation

After obtaining well-curated names, we used an abbreviation generator to produce the abbreviated names. Table 3 summarizes four abbreviation schemes from logical to abbreviated names. We adopted word-level abbreviation and acronym extraction methods and limited word removal and word order change cases, because specifying rules for word removal or order change is an open-ended and difficult-to-scale problem. Our abbreviated name generator employs a probabilistic approach to determine the specific method used for word-level abbreviation. The method is chosen from the following three options, with the selection probability determined by pre-defined weights:

**Method 1** (`keep`): Left as-is. This is trivial but

| Abbreviation Schemes | Examples | |
|---|---|---|
| | Logical Name | Abbreviated Name |
| Word-level Abbreviation | `Current Balance` | `CUR_BAL` |
| Acronym Extraction | `Fiscal Year 2021` | `FY_2021` |
| Word Removal | `Zip Code` | `Zip` |
| Word Order Change | `Birth Rate 2018` | `2018_BR` |

Table 3: Examples of four common abbreviation schemes for logical column names in database tables.

very common, especially when the column header contains fewer words or words that cannot be further shortened without creating ambiguity.

**Method 2** (`lookup`): Replaced with an abbreviation from an expansion-abbreviation look-up table. This is mainly for producing commonly-used abbreviations with enhanced diversity in naming style that is hard to obtain using the above four reformatting methods. Examples include abbreviations based on pronunciations (e.g., `transaction` → `txn` and `end-to-end`→`end2end`), and symbolic conversions (e.g., `second` → `2nd`, `number` → `no./#`, and `at` → `@`). The lookup dictionary contains common abbreviations for 23,110 English words or terms. In cases where multiple candidate abbreviations are available, the abbreviation is chosen randomly.

**Method 3** (`rule`): Processed by one of the word-level abbreviation rules:

**Rule 1**: Keep the first $k$ characters, $k \in [1, 5]$ (e.g. abbreviation $\overset{k=4}{\to}$ abbr);

**Rule 2**: Keep removing non-leading vowels until the threshold $k \in [1, 5]$ or all non-leading vowels have been removed (e.g. abbreviation $\overset{k=5}{\to}$ abbrvtn, doodle $\overset{k=5}{\to}$ doodl);

**Rule 3**: Specifically, while the length of the input string is longer than a specified threshold value $k \in [1, 5]$, the following steps are applied: 1) neighboring duplicate characters are removed, 2) vowels are removed randomly until no vowel remain, and 3) consonants are removed randomly until no consonant remains. (e.g. abbreviation $\overset{k=4}{\to}$ abrv). This is to emulate the real-world data and varying preferences of database developers.

Once a rule is selected from the above choices, it is applied to all words within the same column header. It is important to note that these rules do not apply to non-alphabetical words. In the case of numerical words representing a four-digit year, we shorten it to a two-digit representation with a 50%

**Algorithm 1** Abbreviation generation

**Inputs:**
    A lookup dictionary $\mathcal{D}$ and a string $x$
**Initialize:**
    $\texttt{Method}_\texttt{X} \leftarrow select\_method()$
    $\texttt{Rule}_\texttt{X} \leftarrow select\_rule()$
    $\texttt{abbr.words} \leftarrow \emptyset$
$\mathbf{x} \leftarrow tokenize(x)$          ▷ Input: well-curated
**for** $x_i$ in $\mathbf{x}$ **do**
    **if** $\texttt{Method}_\texttt{X}$ is keep **then**:
        $\tilde{x}_i \leftarrow x_i$
    **if** $\texttt{Method}_\texttt{X}$ is lookup **then**
        **if** $x_i \in \mathcal{D}$ **then**
            $\tilde{x}_i \leftarrow \mathcal{D}[x_i]$
        **else**
            $\tilde{x}_i \leftarrow \texttt{Rule}_\texttt{X}(x_i)$
    **if** $\texttt{Method}_\texttt{X}$ is rule **then**
        $\tilde{x}_i \leftarrow \texttt{Rule}_\texttt{X}(x_i)$
    $\texttt{abbr.words} \xleftarrow{+} \tilde{x}_i$
$\tilde{x} \leftarrow combine(\texttt{abbr.words})$          ▷ Output:
abbreviated

---

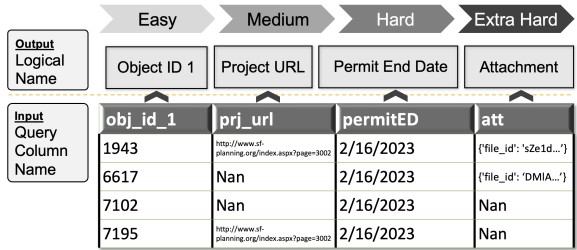

Figure 5: Examples with different difficulty levels. Each example contains a query column name, sampled column contents, and a ground truth logical name.

### 3.3   Evaluation Benchmark Annotation

Instead of splitting a subset of training examples for evaluation, we created a human-annotated evaluation dataset.

#### 3.3.1   Human Annotations

The evaluation dataset was confirmed by 15 human annotators using the City Open Data from Los Angeles, San Francisco, and Chicago. Detailed instructions were provided to the annotators to ensure consistency in annotations. The instructions are outlined as follows:

1. Read the table metadata, including table category, name, and description.

2. Read and understand the original column name and sampled cell values for that column.

3. Determine if the original column name is abbreviated or well-curated. If in a well-curated form, provide only an "abbreviated variant". Otherwise, provide a "well-curated variant" and an "abbreviated variant".

4. When creating abbreviated names, please combine abbreviated words as suggested by the combining rules detailed in Table 3.

A pilot study was first implemented to produce a small number of examples, followed by a quality audit to ensure the guidelines were well-understood. Note that the column names that were found to be unclear or difficult to interpret even with the provided metadata and column cell values were discarded from the dataset. Finally, the annotations underwent another audit process from a separate group of annotators. We employed a criterion where if two out of three annotators agreed, the annotation was considered to pass this agreement measure. Overall, the annotations achieved an agreement rate of 96.5%.

---

probability, e.g., $\texttt{2020} \rightarrow \texttt{20}$.

**Hybrid Method**. To simulate the naming convention for database tables, a probability of $0.5$ is assigned for converting patterns subject to common acronyms in the well-curated column headers. The entire or a part of the well-curated form could be replaced by an acronym, such as $\texttt{Employee Date of Birth} \rightarrow \texttt{EMP\_DOB}$. Furthermore, additional rules are introduced to selectively remove or switch the order of words without altering the semantics of the column name. For example, $\texttt{Event Name} \rightarrow \texttt{Evnt}$ and $\texttt{Mailing Address District 2013} \rightarrow \texttt{2013MailAddrDist}$. Note that when the same word(s) appears in different columns of the same table, we use the same abbreviation form.

**Abbreviation Combination**. As the last step of the abbreviation algorithm, the $combine()$ function assigns an equal probability of concatenating the resulting abbreviated words into *camel case*, *Pascal case*, *snake case*, or *simple combination* which further adds diversity in the naming style.

**Overall Algorithm**. The overall algorithm is in Algorithm 1 with the probability of Method 1 (keep), 2 (lookup) and 3 (rule) set to 0.3, 0.6 and 0.1, respectively. The probabilities for Rule 1, 2, and 3 in rule are set to 0.2, 0.4, and 0.4, respectively. These assignments are undertaken to maintain statistics that resemble real-world datasets.

### 3.3.2 Difficulty Breakdown

We divide the data samples into four difficulty levels to gain deeper insight into the model performance. This classification is based on the edit distance between characters in abbreviated names and ground truth labels. Both names will first be tokenized and have underscores replaced with spaces, and numbers/punctuations were discarded when calculating the edit distance. Four categories are established: (1) 1,036 (11%) easy examples, (2) 3,623 (39%) medium examples, (3) 3,681 (40%) hard examples, and (4) 878 (10%) extra-hard examples. Figure 5 shows one representative example from each level. The difficulty breakdown we employ is significantly different from dataset difficulty breakdown or dataset cartography approaches in literature (Swayamdipta et al., 2020; Ethayarajh et al., 2022), since the column name expansion task is formulated as a generation task as opposed to classification tasks considered in prior literature.

## 4 Methods

Recent advances in pre-trained LMs (Radford et al., 2019; Brown et al., 2020) have shown a strong ability to generate fluent text and the "emergent" performance boost when scaling up LMs (Wei et al., 2022). Therefore, we evaluated the performance of both small and large LMs for NAMEGUESS. We adopted two learning paradigms. First, we employed prompt-based learning with LLMs without tuning model parameters. Second, we fine-tuned small LMs. In particular, we utilized the supervised fine-tuning (Schick and Schütze, 2021a,b) paradigm with task-specific prompts.

**Training**. We fine-tuned pre-trained LMs by contextualizing column query names with table content, incorporating sampled cell values and table schema data. To limit the sequence length of a linearized table, we selected $N$ cell values in the corresponding column (after removing duplicates) for each query name. We truncated cell values when they had more than 20 characters. Moreover, instead of predicting individual query name separately, we jointly predicted $K$ query names. The columns are stacked together to form the table context data $t'$. This structured input is then serialized and combined with a task prompt $q$. Specifically,

- $t' = $ "Column names: $\{x_1\}$, ..., $\{x_K\}$ <SEP> row 1: $\{c_1^1\}$, ..., $\{c_K^1\}$ <SEP> row 2: $\{c_1^2\}$, ..., $\{c_K^2\}$ <SEP> ... row $N$: $\{c_1^N\}$, ..., $\{c_K^N\}$",

- $q = $ "As abbreviations of column names from a table, $\{x_1\}|...|\{x_K\}$ stand for $\{y_1\}|...|\{y_K\}$.".

Here $K$ represents the number of query names, and $N$ is the number of sampled rows in a table. $\{x_i | i = 1, ..., K\}$ refers to the abbreviated column names from the same table. $\{y_i | i = 1, ..., K\}$ represents the ground-truth logical name. And $\{c_i^j | i = 1, ..., K; j = 1, ...N\}$ denotes the sampled cell values of each query name. We set $K$ and $N$ to 10 based on the ablation study results after testing different $K$ and $N$ values in Appendix A.6 Table 7. A table with more than ten columns is split into multiple sequences. The prompt can also be designed in different ways. These prompts were selected by testing various template templates on the GPT-2 XL model and choosing the ones that yielded the best outcomes. We utilized decoder-only models for training, where the entire prompt sequence was employed to recover logical names in an autoregressive manner.

**Prediction**. Given $K$ column names $x_1, ..., x_K$ in a table $t$, we used LM to predict the corresponding logical names $y_1, ..., y_K$. To generate predictions, we concatenated the linearized table context $t'$ and the task prompt $q$ as the input sequence. We used the same query prompt as in training during inference, except that we removed the ground truth. The modified query prompt becomes "As abbreviations of column names from a table, $\{x_1\}|...|\{x_K\}$ stand for". For the non-finetuned LLMs, we provided a single demonstration before the query prompt to ensure that the model can generate answers in a desired format, i.e., "As abbreviations of column names from a table, c_name | pCd | dt stand for Customer Name | Product Code | Date." We extracted the answer in the predicted sequence when the <EOS> token or the period token is met and then split the answers for each query name with the | separator.

## 5 Experiments

We performed a comprehensive set of experiments to answer (1) Can NAMEGUESS be solved as a natural language generation task? (2) How does fine-tuning and scaling the number of parameters help the model handle NAMEGUESS? (3) Can table contents aid disambiguation?

## 5.1 Evaluation Metrics

We use three metrics for evaluation: exact match accuracy, F1 scores based on partial matches, and Bert F1 scores based on semantic matches.

**Exact Match (EM).** Similar to the exact match for question answering, the exact match is computed based on whether the predicted column name is identical to the ground truth after normalization (ignoring cases, removing punctuations and articles).

**F1 Score.** Computed over individual normalized tokens in the prediction against those in the ground truth, as $2 \cdot \text{precision} \cdot \text{recall}/(\text{precision} + \text{recall})$, where precision and recall are computed by the number of shared tokens in the prediction and ground truth label, respectively.

**BertScore.** The similarity score for each token in the predicted phrase with that in the reference phrase computed using pre-trained contextual embeddings (Zhang et al., 2020b). It demonstrates better alignment with human judgment in summarization and paraphrasing tasks than existing sentence-level and system-level evaluation metrics. We use rescaled BertScore F1 and roberta_large (Liu et al., 2019) for contextual embeddings.

## 5.2 Representative Methods

We fine-tuned GPT-2 (Radford et al., 2019) and GPT-Neo (Black et al., 2021) models from Hugging Face. We conducted preliminary experiments using the pre-trained small LMs without fine-tuning but consistently obtained incorrect results. We also evaluated non-finetuned LLMs, including Falcon-40B-Instruct (Almazrouei et al., 2023), LLaMA-65B (Touvron et al., 2023), and GPT-4 (OpenAI, 2023) using the same prompt. Furthermore, we collected human performance from 6 annotators on 1,200 samples from the evaluation set, including 300 from each difficulty level. The annotators had access to 10 sampled cell values for each column. Detailed setups are in Appendix A.5.

## 5.3 Results and Discussion

The main results, including models with and without table context data, are in Table 4. Table 5 reports the EM results on four hardness levels, and the F1 and Bert-F1 results are in Appendix Table 6. The responses of Falcon-instruct, LLaMA, and GPT-4 models may not match the demonstration example, resulting in potential answer extraction failures. In total, 92% of Falcon-instruct-40B, 96% of LLaMA-65B, and 99% of GPT-4 examples

| Model | EM | | F1 | | Bert F1 | |
|---|---|---|---|---|---|---|
| | $q$ | $t'+q$ | $q$ | $t'+q$ | $q$ | $t'+q$ |
| GPT-2 (124M)* | 10.4 | 25.7 | 27.9 | 46.4 | 37.2 | 52.6 |
| GPT-2 (1.5B)* | 35.1 | 37.8 | 56.5 | 59.5 | 61.6 | 63.6 |
| GPT-Neo (1.3B)* | 38.3 | 41.6 | 58.8 | 62.4 | 62.4 | 66.1 |
| GPT-Neo (2.7B)* | 40.6 | **43.8** | 61.4 | 64.7 | 65.7 | **68.2** |
| Human | - | 43.4 | - | **66.5** | - | 65.4 |
| Falcon-Inst. (40B) | 51.2 | 53.4 | 68.4 | 75.6 | 72.2 | 73.3 |
| LLaMA (65B) | 53.3 | 56.2 | 69.6 | 71.2 | 73.3 | 74.0 |
| GPT-4 | 69.3 | **73.4** | 82.2 | **85.5** | 81.6 | **86.2** |

Table 4: Overall exact match (EM), F1, and Bert-F1 scores (%) of fine-tuned models, LLMs (non-finetuned), and human performance. Fine-tuned models are marked with $*$. $t' + q$ indicates the results of incorporating sampled table contents, and $q$ refers to only using task prompts without table contents.

have successfully extracted answers. The scores in the tables are calculated based on the predictions with successful extractions.

**Effect of Model Size.** From Table 4 we see that among the fine-tuned models, the GPT-2-124M model exhibits particularly poor performance, while the fine-tuned GPT-Neo-2.7B model achieves the highest overall EM, F1, and Bert F1 scores. With a similar number of parameters, the GPT-Neo-1.3B model has a 3.8% higher overall EM score than GPT-2-1.5B. With a much larger size, GPT4 achieves 29.6% higher EM than the best fine-tuned model.

**Effect of Fine-Tuning.** Without fine-tuning, the small to medium models achieve <1% overall EM scores, producing almost random predictions. However, the NAMEGUESS training data allows a series of fine-tuned small to medium-sized models to approach human performance with a much cheaper inference cost than LLMs. Still, a big gap exists between these fine-tuned models and LLMs.

**Human Performance.** One important observation is that human performance on NAMEGUESS test set (also human-annotated) is far from perfect. However, the fine-tuned models can slightly exceed human performance (the fine-tuned GPT-Neo-2.7B is 0.4% higher in EM). Intuitively, expanding from query name $x$ into logical name $y$ is much more challenging than reverse since it requires a deeper understanding of the meaning and context of the abbreviations to identify and reconstruct the original phrase accurately.

**Effect of table context data.** Expanding abbreviated column names may require a deep understanding of table content. For example, "E_NAME" can represent "Employer Name" instead of "Employee

| | **Easy** | **Med-ium** | **Hard** | **Extra Hard** |
|---|---|---|---|---|
| GPT-2 (124M) * | 41.4 | 30.3 | 19.0 | 16.1 |
| GPT-2 (1.5B) * | 53.8 | 42.7 | 32.3 | 21.1 |
| GPT-Neo (1.3B) * | 54.3 | 47.2 | 36.8 | 23.6 |
| GPT-Neo (2.7B) * | 55.3 | 50.4 | 38.0 | 27.2 |
| Falcon-Inst. (40B) | 62.2 | 59.3 | 49.0 | 36.5 |
| LLaMA (65B) | 62.2 | 62.9 | 52.3 | 37.9 |
| GPT-4 | 71.7 | 79.9 | 72.0 | 54.7 |
| Human | 53.7 | 53.7 | 33.7 | 30.2 |

Table 5: Exact match scores (%) of fine-tuned models, LLMs (non-finetuned), and human performance for four difficulty levels. * indicates fine-tuned models.

Name" with company names among column values. By comparing the performance of fine-tuned and non-fine-tuned models with sampled table content ($t' + q$) and without table content ($q$ only) in Table 4, we find that incorporating sampled table contents can increase the performance of all the models, with a huge boost of 15% for the smallest GPT-2 (124M) model.

**Difficulty Breakdowns.** We observe a trend of decreasing exact match scores on more difficult examples. From the difficulty breakdown, the fine-tuned small LMs can outperform humans on easy examples but are worse on extra hard examples. Conversely, GPT-4 outperforms all other models and human results, especially in medium and hard divisions. The LLMs are infused with broader knowledge regarding hard and extra hard examples. They are better at interpreting the exact meaning of the misleading abbreviations that often involve uncommon acronyms words or ambiguous ways of combining abbreviations.

# 6 Related Work

## 6.1 Table Understanding and Metadata Augmentation Tasks

Great strides have been achieved in table understanding (Dong et al., 2022). Descriptive column names are crucial for task performance, aiding the model in comprehending table semantics and conducting cell-level reasoning. Column type detection (Hulsebos et al., 2019; Zhang et al., 2020a; Suhara et al., 2022) and column relationship identification (Deng et al., 2022; Iida et al., 2021; Wang et al., 2021) involve assigning predefined types, like semantic labels or database column relationships. Semantic column type detection and NAMEGUESS, though related to columns, have distinct objectives. The former predicts types with

predefined labels (classification task), while the latter refines tokens within names, often at the word level (e.g., "c_name" expands to "customer name"). Table question answering (Yin et al., 2020; Herzig et al., 2020; Yang et al., 2022; Xie et al., 2022) requires models to understand both tables and natural language questions and to perform reasoning over the tables. Other tasks, such as table description generation (Gong et al., 2020), table fact verification (Eisenschlos et al., 2021), and formula prediction (Cheng et al., 2022), also require meaningful column names to aid the model in understanding the overall semantic meaning of tables.

## 6.2 Abbreviation Expansion and Acronym Disambiguation

Abbreviations are widely used in social network posts (Gorman et al., 2021), biomedical articles (Yu et al., 2002), clinic notes (Wu et al., 2015) and scientific documents (Zilio et al., 2022; Pouran Ben Veyseh et al., 2020). Abbreviation expansion and acronym disambiguation tasks are typically formulated as classification problems (Ammar et al., 2011; Pouran Ben Veyseh et al., 2020), which involve selecting an expansion for an abbreviation from a set of candidates based on the context information. The ad hoc abbreviation expansion task (Gorman et al., 2021) is similar to the NAMEGUESS task but limits the per-token translation imposed on the dataset creation process and the developed solutions. Meanwhile, NAMEGUESS is a natural language generation problem suitable for a wide range of lengths of abbreviations and expansions. Regarding abbreviation expansion, our most relevant work is by Cai et al. (2022). However, this work primarily addresses text messages/SMS abbreviations, aiming to reduce message length and minimize typos. In contrast, our task focuses on generating meaningful and human-readable expansions for column name-based abbreviations.

# 7 Conclusion and Future Works

We introduced a new task related to expanding the commonly-used abbreviated column names in tabular data, developed and created two benchmark datasets to facilitate the study of this task, and analyzed the performance of many language modeling methods to solve this task. One future direction is to utilize similar examples that provide contextual information to solve this task, preferably feeding these examples through in-context learning.

## 8 Ethics Statement

The human annotations, including abbreviated/logical column names for the evaluation set, were collected through hired annotators from a data annotation service. Annotators were instructed to strictly refrain from including any biased, hateful, or offensive content towards any race, gender, sex, or religion. The annotations passed through audits, where they were examined by a separate group of annotators and reached a 96.5% agreement ratio. The human performance on the NAMEGUESS test set was collected from database and dialogue/linguistics experts.

## 9 Limitations

One limitation of our work is the need for real relational database tables. The training and evaluation sets we used were all public web-related tables, which generally have fewer rows and lack the metadata of primary and secondary keys. This work is just the first step to introduce this topic to the NLP community, and further research is needed to improve the performance with methods that better capture context around relational data. Moreover, the scope of handling omitted information in the original column names, like column header "glucose", which stands for "fasting glucose", is beyond NAMEGUESS. One possible solution is to collect and utilize more table metadata information. For example, if the table name contains the term "fasting", then the original column name "glucose" will most likely be inferred as "fasting glucose".

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

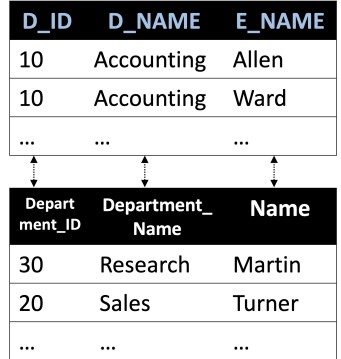

(a) An example of unionable relation extraction task.

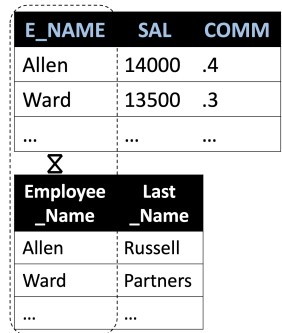

(b) An example of joinable relation extraction task.

Figure 6: Unionable/Joinable relation extraction tasks with abbreviated column names. The column names in blue are the corrupted names. Column `D_ID` should match `Department_ID`, `D_NAME` with `Department_Name`, and `E_NAME` with `Name` for unionable column pairs. Column `E_NAME` matches with column `Employee_Name` for the joinable relation.

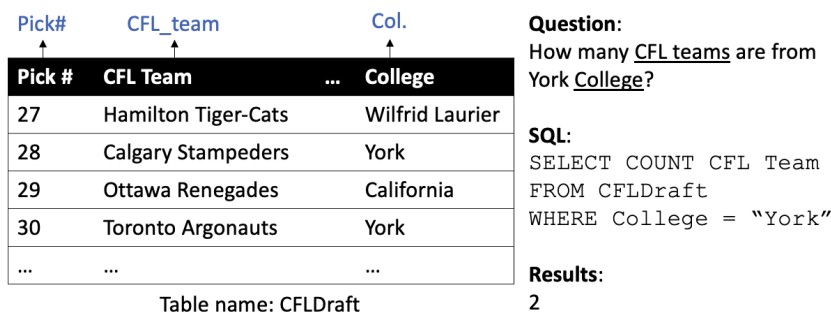

**Question**:
How many CFL teams are from York College?

**SQL**:
```
SELECT COUNT CFL Team
FROM CFLDraft
WHERE College = "York"
```

**Results**:
2

Figure 7: An table Question Answer (QA) task example. After corrupting the column names in the table (i.e., the names in blue), the keyword in the question 'college' may fail to locate the abbreviated column name `Col.`.

## A    Appendix

### A.1    The Effect of Abbreviated Column Names on Table Understanding Tasks

**Unionable and Joinable Relation Detection**. The unionable and joinable column relation extraction tasks are illustrated in Figure 6. The original recall score of the schema-based relation detection (with expanded column name) in Table 1 is cited from the Valentine paper (Koutras et al., 2021). The datasets used in Valentine were fabricated by corrupting the column names or the column cells. The results of the original datasets have the same column names in two tables, so the recall score is 1.0. The best schema-based method (i.e., *Similarity Flooding* (Melnik et al., 2002)) drops from 1.0/1.0 to 0.63/0.56 (an average score of 0.595) on unionable/joinable relation detection tasks with expanded and abbreviated column names. As illustrated in the Valentine study, even when other schema information, such as data types and transi-tive relationships, is available, the lack of descriptive column names can hinder the effectiveness of schema-based methods in identifying unionable and joinable columns.

**Table Question Answering** Table question answering (QA) aims to derive answers from tables for natural language questions. Typically, the task requires two steps: translating natural language questions to corresponding SQL queries and then extracting the answers from the SQL queries. In order to test the effect of the abbreviations of the column names for Table QA, we used the abbreviated method in section 3.2.2 to corrupt the column names in tables of the WikiSQL (Zhong et al., 2017) dataset and used the same training and evaluation script in UnifiedSKG (Xie et al., 2022) with the T5-large model. The accuracy drops from 84.32 to 80.49, recorded in Table 1.

## A.2 The Logical Name Identification Algorithm

See Algorithm 12.

---
**Algorithm 2** Logical name identification
---
**procedure** IS_LOGICAL_NAME($x$)
  **if** $x \in \mathcal{V}$ **then**
    **return** 1                                 ▷ well-curated
  **else**
    $\mathbf{x} \leftarrow tokenize(x)$
    **for** $x_i \in \mathbf{x}$ **do**
      **if** $isdigit(x_i)$ **then**:
        **continue**
      $x_i \leftarrow lemmatize(x_i)$
      **if** $x_i \notin \mathcal{V}$ **then**
        **return** 0                             ▷ not well-curated
    **return** 1                                 ▷ well-curated
---

## A.3 Training/Evaluation Set Details

Some variations in the table size are included in the training and evaluation set. As seen from Figure 8, most tables have fewer than 100 columns, and there could be some small tables with fewer than five columns since the columns that initially had abbreviated ambiguous names were filtered before generating abbreviated columns. We limit the number of rows per table for training and evaluation sets to less than 1000.

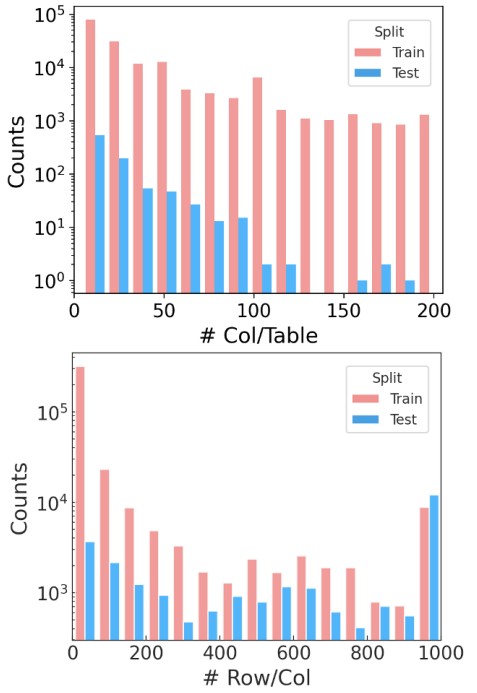

Figure 8: Additional statistics for NAMEGUESS training and test datasets. Top: The distribution of the number of columns per table. Bottom: The distribution of the number of rows (cells) per column.

## A.4 Annotation Interface

Figure 9 shows the interface that the annotators used to produce both the abbreviated column names ("Cryptic Variant") and well-curated column names ("Well-Curated Variant") based on the column contents ("Original Column Name", "Sampled Column Values") and table metadata ("Table Name", "Table Category", and "Table Description"). The "Comments" column includes some explanations or clarifications for ambiguous samples.

## A.5 Experimental Setup

All fine-tuning experiments on GPT-2[1], GPT-2-XL[2], GPT-Neo-1.3B[3], GPT-Neo-2.7B[4] and inferences were run using eight NVIDIA A100 GPUs. The inference results from GPT-4 was generated using OpenAI API calls[5]. The model weights for LLaMA-65B[6] and Falcon-Instruct-40B [7] models were downloaded and hosted using Text Generation Inference (TGI) toolkit [8]. We used a global batch size of 128 and a learning rate of $10^{-6}$ during the fine-tuning process. We used beam search with a beam size of 5 during inference, which performed better than decoding by sampling. The results of the evaluation benchmark reported in the experimental tables were obtained from the best checkpoint of each model on a single run.

---

[1]https://huggingface.co/gpt2
[2]https://huggingface.co/gpt2-xl
[3]https://huggingface.co/EleutherAI/gpt-neo-1.3B
[4]https://huggingface.co/EleutherAI/gpt-neo-2.7B
[5]https://platform.openai.com/docs/models/gpt-4
[6]https://ai.meta.com/blog/large-language-model-llama-meta-ai/
[7]https://huggingface.co/tiiuae/falcon-40b-instruct
[8]https://github.com/huggingface/text-generation-inference

| | Easy | | | Medium | | | Hard | | | Extra Hard | | |
|---|---|---|---|---|---|---|---|---|---|---|---|---|
| Model | EM | F1 | B-F1 | EM | F1 | B-F1 | EM | F1 | B-F1 | EM | F1 | B-F1 |
| GPT-2 (124M) * | 41.4 | 65.2 | 66.9 | 30.3 | 44.9 | 54.4 | 19.0 | 44.4 | 49.2 | 16.1 | 38.4 | 42.8 |
| GPT-2 (1.5B) * | 53.8 | 73.4 | 73.7 | 42.7 | 58.7 | 65.5 | 32.3 | 59.4 | 62.1 | 21.1 | 46.8 | 50.2 |
| GPT-Neo (1.3B) * | 54.3 | 72.8 | 74.4 | 47.2 | 62.4 | 68.5 | 36.8 | 62.9 | 64.8 | 23.6 | 48.1 | 51.6 |
| GPT-Neo (2.7B) * | 55.3 | 73.8 | 76.5 | 50.4 | 64.7 | 70.7 | 38.0 | 64.8 | 66.4 | 27.2 | 53.0 | 56.0 |
| Falcon-Inst. (40B) | 62.2 | 75.6 | 77.2 | 59.3 | 71.2 | 76.0 | 49.0 | 71.1 | 72.2 | 36.5 | 58.2 | 62.0 |
| LLaMA (65B) | 62.2 | 73.9 | 77.2 | 62.9 | 72.1 | 76.6 | 52.3 | 72.7 | 73.6 | 37.9 | 57.9 | 60.9 |
| GPT-4 | 71.7 | 81.1 | 81.2 | 79.9 | 87.6 | 89.5 | 72.0 | 87.3 | 86.8 | 54.7 | 74.4 | 75.7 |
| Human | 53.7 | 74.4 | 74.8 | 53.7 | 69.3 | 72.8 | 33.7 | 64.7 | 59.9 | 30.2 | 54.4 | 48.3 |

Table 6: Exact match (EM), F1, and BertScore-F1 (B-F1) scores (%) of fine-tuned models, LLMs (non-finetuned), and human performance for four difficulty levels. * Fine-tuned models.

### A.6 Ablation Studies

#### A.6.1 Difficulty Breakdowns.

The full results of all the models on four different difficulty levels are in Table 6.

#### A.6.2 The Effect of Different Jointly Predicted Query Column Names $K$ and Different Number of Sampled Rows $N$.

We evaluated the exact match (EM) scores on varied values of $K$ and $N$ on the LLaMA-65B model. From Table 7, we note that an increase in the values of $K$ and $N$ results in a rise in EM scores, although the rate of increase tends to diminish for higher $K$ and $N$ values. Thus, we set the values of $K$ and $N$ at 10 for other models in the experiments.

| #column name($K$) / #sampled row ($N$) | N=1 | N=10 |
|---|---|---|
| K=1 | 45.7 | 49.0 |
| K=5 | 54.3 | 55.5 |
| K=10 | **56.0** | **56.2** |

Table 7: Exact match scores (%) with different $K$ and $N$ values for the LLaMA-65B model.

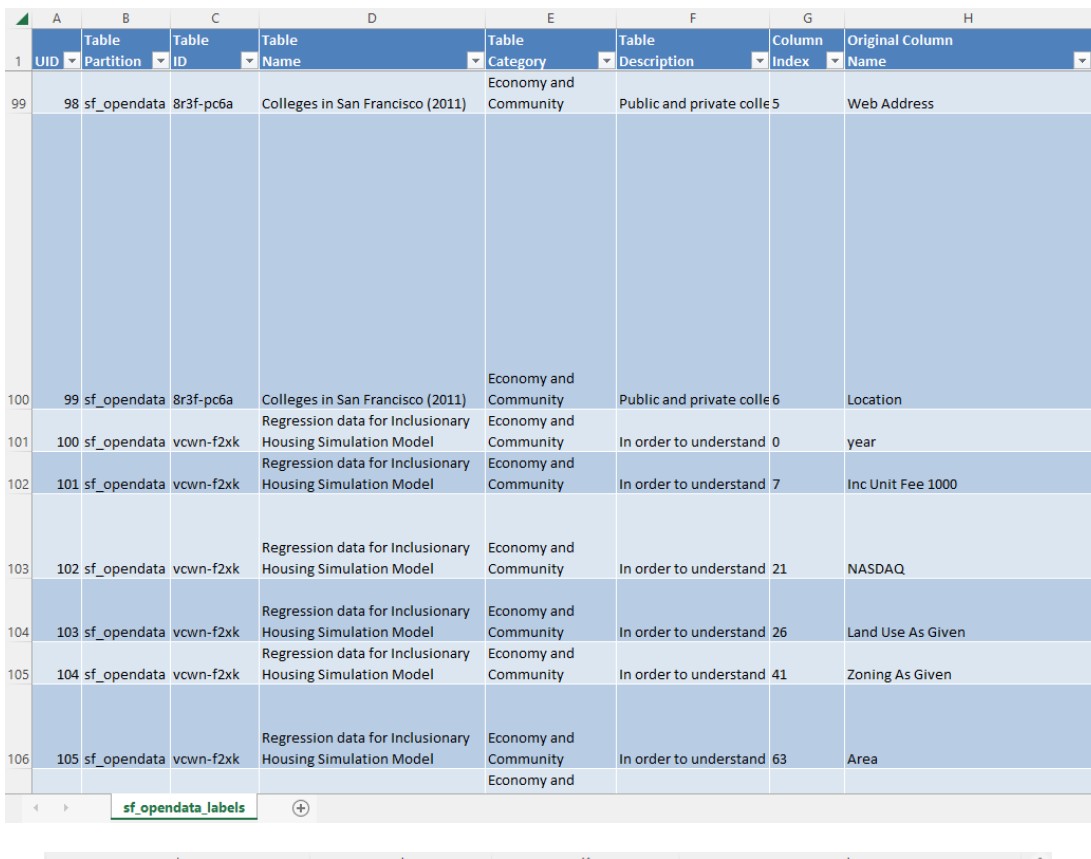

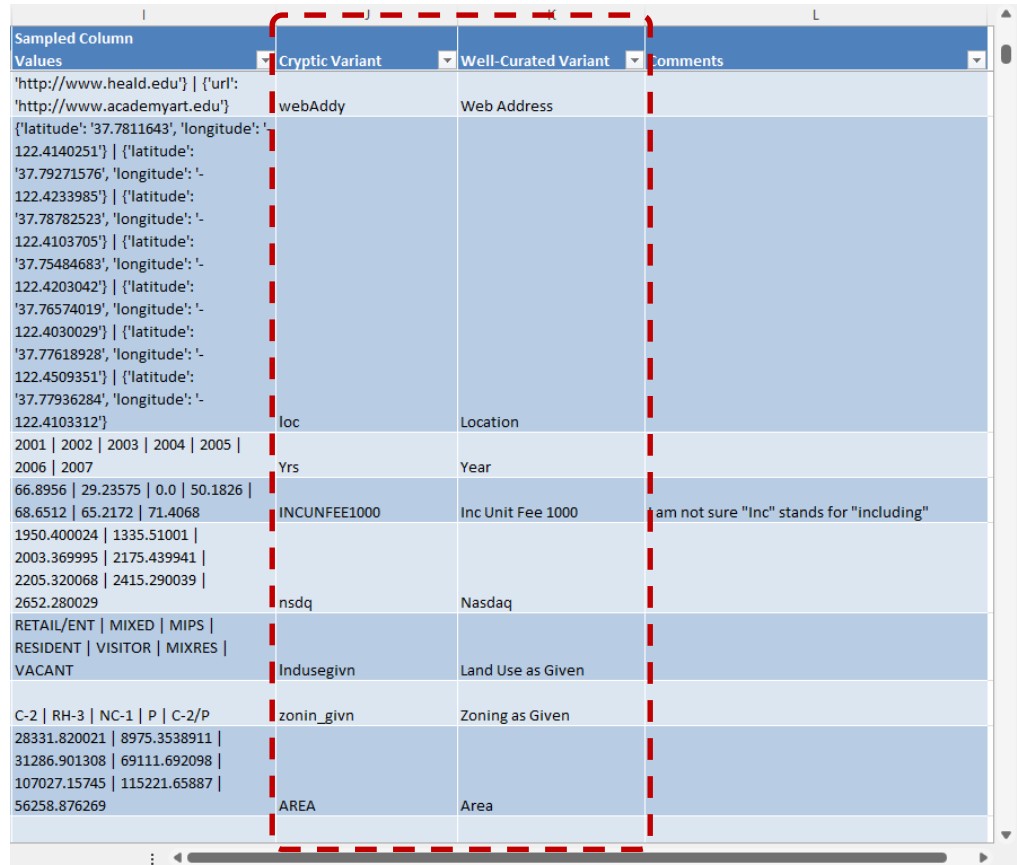

Figure 9: A screenshot for the interface used by annotators.