# OpenReview forum: "NameGuess: Column Name Expansion for Tabular Data"
_EMNLP/2023/Conference — EMNLP 2023 Main_

### Official Review · Reviewer_hjkH · 2023-07-27

**Typos Grammar Style And Presentation Improvements:** None
**Soundness:** 4

**Excitement:**

4: Strong: This paper deepens the understanding of some phenomenon or lowers the barriers to an existing research direction.

**Missing References:**

None

**Paper Topic And Main Contributions:**

In this paper, the authors have introduced a new task called NAMEGUESS, which tackles expanding abbreviated column names in tabular data. Abbreviating column names is common while building tables within standard database systems; it is convenient for representation and use in code. However, it negatively impacts understandability and causes problems for downstream algorithmic tasks, such as text2SQL, TableQA, Column Relation discovery, etc. The authors have empirically demonstrated this assertion via experiments in Table 1, thereby firmly stating why this task is essential.

The authors introduce NAMEGUESS as a natural language task, unlike existing textual abbreviation expansion tasks, which tend to use a classification-style approach via a predefined set of candidate expansions. This makes NAMEGUESS a more challenging task that is much more aligned to the real world use-cases. For empirical evaluations, the authors have built a large-scale training/test dataset for this task.

For building the training dataset, the authors took the expanded column names as input and used the convention of the database developers to convert the expanded column names into abbreviated names. The tables were collected from seven public tabular dataset repositories (Sec 3.1). An expanded column name from a table is regarded as well-curated if all the tokens in this column name were found in a predefined vocabulary. Only well-curated expanded column names are converted into abbreviated names. The abbreviation generation step is largely rule-based, consisting of four primary schemes: Word-level abbreviation, Acronym extraction, word removal, and word-order change. This procedure yields 384,333 columns across 163,474 tables.

A human-annotated evaluation dataset was built using 15 human annotators, wherein the annotators were asked to create an "abbreviated variant" for all the "well-curated" original column names using similar rules as the abbreviation generation step above. This step yields an evaluation benchmark consisting of 9,218 examples across 8,965 tables.

The authors compare the performances of small and large Language models (LMs) for this task in a zero-shot/finetuned setting. For finetuning, the authors finetuned the task-specific prompts. The training procedure was done using decoder-only models in an autoregressive manner. The model was queried for prediction until either an <EOS> or period token was returned. The authors analyzed different LMs on this task across different paradigms, such as model sizes, effect-of-finetuning, significance of table context data, etc., and provided a comprehensive analysis of various model performances.

**Questions For The Authors:**

1. The evaluation setup consists of decoder-only models; How do the encoder-decoder models, for e.g., let's say flan/t5-* family of models, perform on this task?

**Reasons To Accept:**

1. The significance of the NAMEGUESS task has been properly motivated. It is an essential task at the intersection of NLP and Database communities, which could have exciting implications within enterprise settings, wherein data assets typically sit within SQL databases, and their metadata, for example, column headers, are generally abbreviated.

2. The paper is well-written and easy-to-follow. The analysis of the results and the ablation studies provide a complete picture of how the results vary with changing hyper-parameters/table contexts. Additionally, the benchmark dataset, if made publicly available, would engage and motivate researchers to build improved models on this task.


**Reasons To Reject:**

None

**Reproducibility:**

4: Could mostly reproduce the results, but there may be some variation because of sample variance or minor variations in their interpretation of the protocol or method.

**Reviewer Confidence:**

4: Quite sure. I tried to check the important points carefully. It's unlikely, though conceivable, that I missed something that should affect my ratings.

---

> ### Author Rebuttal · Authors · 2023-08-27
>
> We appreciate the insightful feedback from the reviewer, which highlights a pertinent aspect of our evaluation setup.
>
> 1. **Evaluation Setup**: It's worth noting that our current access is limited to large-scale out-of-the-box decoder-only models, like GPT-based models, Falcon, and LLaMA models. As a result, we fine-tuned smaller decoder-only models to maintain a balanced and fair comparison. We recognize the significance of this aspect and plan to explore and incorporate more models, including encoder-decoder variants, in future research endeavors. This expansion will provide a broader understanding of model performance across different architectures.

---

### Official Review · Reviewer_4P1D · 2023-07-28

**Soundness:** 4

**Excitement:**

3: Ambivalent: It has merits (e.g., it reports state-of-the-art results, the idea is nice), but there are key weaknesses (e.g., it describes incremental work), and it can significantly benefit from another round of revision. However, I won't object to accepting it if my co-reviewers champion it.

**Missing References:**

Yoshihiko Suhara, Jinfeng Li, Yuliang Li, Dan Zhang, Çagatay Demiralp, Chen Chen, Wang-Chiew Tan:
Annotating Columns with Pre-trained Language Models. SIGMOD Conference 2022: 1493-1503

Dan Zhang, Yoshihiko Suhara, Jinfeng Li, Madelon Hulsebos, Çagatay Demiralp, Wang-Chiew Tan:
Sato: Contextual Semantic Type Detection in Tables. Proc. VLDB Endow. 13(11): 1835-1848 (2020)

Enzo Veltri, Gilbert Badaro, Mohammed Saeed, Paolo Papotti:
Data Ambiguity Profiling for the Generation of Training Examples. ICDE 2023: 450-463

**Paper Topic And Main Contributions:**

This paper is about naming conventions in tabular data, specifically focusing on the challenge of abbreviated column headers that can cause confusion and impact data comprehension and accessibility. It introduces a new task, called NameGuess, which aims to expand abbreviated column headers into full names, formulating it as a natural language generation problem.

The main contributions are
1. New task modeled as natural language generation problem.
2. A data generation method that produces a large-scale training set consisting of 384K abbreviated-expanded column pairs.
3. Human-annotated evaluation benchmark with 9.2K examples drawn from real tables. This benchmark spans various difficulty levels, providing a reference for comparing the effectiveness of various solutions.
4. Comprehensive evaluation of LLMs of varying sizes and training strategies, including comparison to human performance.

**Questions For The Authors:**

Q1.
Why the partition of the datasets with city open data only as test data?

Q2.
There are several choices that lead to a very well curated dataset, which may not reflect the reality for datasets in the wild.
Why constraining the human annotation process to the rules of the generator? (point 4 in 3.3.1)
Why col names unclear or difficult have been removed? (again 3.3.1) this makes the dataset much easier and far away from reality

Q3
Sec 3 states 8.9 tables, but  table 2 and fig 4 states 895 - which number is correct?

Q4
why reporting only the predictions with successful extractions (sec 5.2)? this seems to possibly favor LLMs - are these examples removed also from the other models? I d argue that LLMs "fail" in this case and should be reported

**Reasons To Accept:**

A1
Pertinent and practical problem in dealing with tabular data.
The approach of treating column name expansion as a natural language generation issue is sound.

A2
The creation of a large-scale training set and a human-annotated benchmark is a valuable contribution. The creation process shows rigor and understanding of the challenges in expanding abbreviated column names (abbreviation ambiguity, developer conventions, and domain-specific terminologies).

A3
The evaluation of LLM against human performance on the NameGuess task is a good first reference for future research for this task.

**Reasons To Reject:**

R1.
Very limited coverage of work done on profiling tables to obtain metadata.
The paper should better position the work
While the mentioned work (including turl) are not immediate baselines, it is not clear why tabular language models cannot be used immediately as a solution for the proposed task. Other related work include

Yoshihiko Suhara, Jinfeng Li, Yuliang Li, Dan Zhang, Çagatay Demiralp, Chen Chen, Wang-Chiew Tan:
Annotating Columns with Pre-trained Language Models. SIGMOD Conference 2022: 1493-1503

Dan Zhang, Yoshihiko Suhara, Jinfeng Li, Madelon Hulsebos, Çagatay Demiralp, Wang-Chiew Tan:
Sato: Contextual Semantic Type Detection in Tables. Proc. VLDB Endow. 13(11): 1835-1848 (2020)

Enzo Veltri, Gilbert Badaro, Mohammed Saeed, Paolo Papotti:
Data Ambiguity Profiling for the Generation of Training Examples. ICDE 2023: 450-463

notice how the last one covers the ambiguity problem that is mentioned in the submission but not really tackled.

R2.
The task is heavily dependent on training data generated using abbreviation lookups and probabilistic rules, i.e., with a synthetic approach. These rules might not scale well with strings that do not adhere to common abbreviation patterns or for languages other than the one studied.

R3.
Some important decisions in the study are not clear and require justifications. My major concerns are
(i) the abbreviations in the test datasets are designed to be consistent with the abbreviation created for the synth training data;
(ii) datasets have been polished of hard examples, thus widening the gap with real ones (see also R2)

**Reproducibility:**

4: Could mostly reproduce the results, but there may be some variation because of sample variance or minor variations in their interpretation of the protocol or method.

**Reviewer Confidence:**

4: Quite sure. I tried to check the important points carefully. It's unlikely, though conceivable, that I missed something that should affect my ratings.

**Typos Grammar Style And Presentation Improvements:**

- in 3.3.1, clarify what is the measured agreement: is this about giving the same abbreviation for a given column?

- it is not clear how the probabilities in 3.2.2 are assigned

- in 3.1, "Half of the column names being duplicates" - how is this possible? all col names should be different, in general

---

> ### Author Rebuttal · Authors · 2023-08-27
>
> We appreciate Reviewer 2's insightful feedback to enhance the clarity and accuracy of our work. We have carefully considered each of your questions and concerns and would like to address them accordingly.
>
> Q1. **Choice of Test Data Partition**: The decision to use part of the city open data as the test set is motivated by the practical constraints of human annotation costs. The choice of city open data as the test set was primarily driven by its larger table size and more diverse column headers, which makes it a suitable representative for evaluation.
>
> Q2. **About the Guidance of Human Annotation**: 1) For the evaluation datasets, we have kept both well-curated and abbreviated column names to capture the realistic scenarios encountered with real-world datasets. To make this clear, we will edit Point 3 of Section 3.3.1 as "*Determine if the original column name is abbreviated or well-curated. If in a well-curated form, annotators are asked to provide an "abbreviated variant". Otherwise, annotators will add both a "well-curated variant" and an "abbreviated variant" to the benchmark*." 2) With Point 4 of Section 3.3.1 "*When creating the abbreviated names, the abbreviated words can be combined using any of the combining rules defined for the abbreviation generator*", we will clarify that it serves as a guidance rather than a strict limitation on how annotators should combine abbreviated words. This guidance is to ensure the diversity of annotations like "*word-level abbreviation*" and "*acronym extraction*". 3) We did not remove unclear or difficult abbreviated column names, but the annotations. To make this clear, we will edit the sentence "*Note that the annotated column names that ... were discarded from the dataset.*".
>
> Q3. **Inconsistency in Table and Section Numbers**: The accurate number of tables in the evaluation dataset is 895, as reflected in Table 2 and Figure 4. We will fix this in the final version of the paper.
>
> Q4. **Reporting Predictions with All Examples**: We understand your point and agree that it is crucial to provide a more comprehensive view of the model performance. To address this, we will augment the results in Table 4 by including, for example, the exact-match scores under $t'+q$ for Falcon-Inst.(40B), LLaMA(65B), and GPT-4, which are 49.1%, 54.0%, and 72.7%, respectively. This adjustment will provide a more balanced perspective on the performance of all models.
>
>
>
> **Presentation Improvements**:
>
> 1) in 3.3.1, clarify what is the measured agreement: is this about giving the same abbreviation for a given column?
>
> - **Answer**: We had a separate group of annotators to do the quality audit. To determine if an annotation is reasonable, we employed a criterion where, if two out of three annotators are in agreement, the annotation is considered to pass this agreement measure.
>
> 2. how the probabilities in 3.2.2 are assigned?
>
> - **Answer**: The probabilities assigned in section 3.2.2 are pre-assigned by the developers. This assignment is undertaken to maintain statistics that resemble those found in real-world datasets. By incorporating these probabilities, we strive to create a synthetic dataset that retains realistic characteristics, thus enhancing the applicability and relevance of our approach.
>
> 3. in 3.1, "Half of the column names being duplicates" - how is this possible? all col names should be different, in general.
>
> - **Answer**: The reference to "Half of the column names being duplicates" pertains to the potential occurrence of identical column names across different tables. In real-world scenarios, certain column names, such as "customer id," might appear repeatedly in various tables to signify the same type of information. To clarify, we do not mean that the column names within a single table are duplicated, but rather that identical column names may occur in different tables due to the nature of the data being represented.

---

### Official Review · Reviewer_MEbZ · 2023-08-09

**Typos Grammar Style And Presentation Improvements:** N/A.
**Soundness:** 4

**Ethical Concerns:**

Yes

**Excitement:**

4: Strong: This paper deepens the understanding of some phenomenon or lowers the barriers to an existing research direction.

**Justification For Ethical Concerns:**

N/A.

**Missing References:**

N/A.

**Paper Topic And Main Contributions:**

In this work, authors introduced an innovative NLP task known as "column name expansion," along with a collection of datasets that encompasses human evaluations, data difficulty breakdown, and benchmark evaluations. In this paper, the detailed data collection steps, difficulty breakdown, logical name identification, abbreviation generation, human annotations are clearly described.

**Questions For The Authors:**

See above.

**Reasons To Accept:**

- I am always greatly appreciating papers that give a presentation of new tasks, and provide the provision of new datasets and evaluate benchmarks. This is the primary reason for my positive comments (constituting over 90%), as such actions infuse new vitality into the field! Clearly, this paper deserves an acceptance.

- The details are all clear, and the results look credible.

**Reasons To Reject:**

I have few concerns that should be considered / or explained (e.g., in the limitation section):

- In practical work, the challenge with abbreviated column names in tables often lies not in the recognization of an abbreviation, but rather in dealing with omitted information. For instance, in diabetes datasets, "glucose" might refer to random glucose, fasting glucose, and so on. Similarly, measurement methods can exhibit considerable variation due to different techniques being employed. Shouldn't we be focusing more on addressing these kinds of issues? Of course, I'm not a nitpicking reviewer – this paper is already quite good. However, I still encourage you to provide some explanations and analyses.

- From my experience, LLM embeddings have shown good understanding for abbreviations. I'm not sure if the presented abbreviation cases are not particularly challenging or important (GPT-4 could easily handle this issue, and I personally think the extra hard case you give is not very difficult). Could you showcase and analyze how, through column name expansion, we can achieve significant gains in some certain NLP tasks? Maybe an additional experiment is required.

**Reproducibility:**

5: Could easily reproduce the results.

**Reviewer Confidence:**

4: Quite sure. I tried to check the important points carefully. It's unlikely, though conceivable, that I missed something that should affect my ratings.

---

> ### Author Rebuttal · Authors · 2023-08-27
>
> We appreciate the thoughtful feedback and suggestions on our paper. We have carefully considered the points raised and would like to address them in our research.
>
> - **Handling Omitted Information**:
> Reviewer 1 has raised a valid concern regarding the challenge of dealing with omitted information when using abbreviated column names in practical applications. We agree that this is an important aspect to consider. Our current problem scope does not directly address the semantic meaning expansion problem. We will acknowledge this limitation in the paper. Moreover, as outlined in the *future works* section, we believe that the incorporation of a codebook that associates abbreviations with their expanded meanings (including the omitted information) could mitigate the omitted information issue. As suggested in the feedback, if “glucose” stands for “fasting glucose”, then we can use this example as an in-context demonstration to help interpret similar terms. Another possible solution is to collect and utilize more table metadata information, for example if the table name contains the term “fasting”, then the original column name ‘glucose’ is most likely to be inferred as “fasting glucose”.
>
> - **Performance Gain in NLP tasks with Expanded Column Names**:
> We appreciate Reviewer 1's insights into the capabilities of LLMs in understanding abbreviations. We have compared the performance of using abbreviated and expanded column names on three table understanding tasks, i.e., text2SQL, schema-based relation detection, and table QA in Table 1 of the *introduction* section. The performance drops on all the tasks when they have abbreviated column names. These results can show the benefits of the expanded column names for these NLP tasks.

---

### Meta-Review · Area_Chair_E1Y5 · 2023-09-19

**Recommendation:** 5

**Metareview:**

The reviewers are positive about the paper. They find the task well-motivated and the paper well-written. All reviewers rate the soundness of the paper high and two of the three are highly excited to see the paper published (the third is ambivalent). The reviewers raised issues that seem easy to fix before the camera-ready. All reviewers consider the paper reproducible.

---

### Decision · Program_Chairs · 2023-10-07

**Decision:**

Accept-Main

**Comment:**

The reviewers are positive about the paper. They find the task well-motivated and the paper well-written. All reviewers rate the soundness of the paper high and two of the three are highly excited to see the paper published (the third is ambivalent). The reviewers raised issues that seem easy to fix before the camera-ready. All reviewers consider the paper reproducible.